# Disorders of the Calcium Sensing Signaling Pathway: From Familial Hypocalciuric Hypercalcemia (FHH) to Life Threatening Conditions in Infancy

**DOI:** 10.3390/jcm11092595

**Published:** 2022-05-05

**Authors:** Jakob Höppner, Kathrin Sinningen, Adalbert Raimann, Barbara Obermayer-Pietsch, Corinna Grasemann

**Affiliations:** 1Department of Pediatrics, St. Josef-Hospital Bochum, Ruhr-University Bochum, 44791 Bochum, Germany; jakob.hoeppner@uni-wh.de (J.H.); kathrin.sinningen@ruhr-uni-bochum.de (K.S.); 2Center of Expertise for Rare Disorders of Bone, Growth and Mineralization, Vienna Bone and Growth Center, 1090 Vienna, Austria; adalbert.raimann@meduniwien.ac.at; 3Endocrinology Lab Platform, Division of Endocrinology and Diabetology, Department of Internal Medicine and Department of Gynecology and Obstetrics, Medical University Graz, 8036 Graz, Austria; barbara.obermayer@medunigraz.at

**Keywords:** familial hypocalciuric hypercalcemia (FHH), calcium sensing receptor (*CaSR*), neonatal hyperparathyroidism (NHPT), neonatal severe hyperparathyroidism (NSHPT), cinacalcet

## Abstract

Familial hypocalciuric hypercalcemia (FHH) is a mostly benign condition of elevated calcium and PTH levels based on a hyposensitive calcium sensing receptor (*CaSR*) in FHH 1 or its downstream regulatory pathway in FHH2 and FHH3. In children, adolescents and young adults with FHH the main challenge is to distinguish the condition from primary hyperparathyroidism and thereby to avoid unnecessary treatments including parathyroidectomy. However, inheritance of FHH may result in neonatal hyperparathyroidism (NHPT) or neonatal severe hyperparathyroidism (NSHPT), conditions with high morbidity, and in the latter even high mortality. This review focuses on the genetic and pathophysiological framework that leads to the severe neonatal form, gives recommendations for counselling and summarizes treatment options.

## 1. Introduction

Loss-of-function (LOF) mutations in the calcium-sensing pathway may cause various calcium-hyposensitivity-diseases which are characterized by elevated serum calcium levels. Pathophysiologically, this is based on a shift of the ‘set point’ of the calcium concentration at which PTH secretion is half-maximal [1]. The resulting disorders share a constellation of inappropriately high PTH concentrations despite elevated serum calcium levels and are categorized into three groups based on genetic origin, time of onset and severity of clinical symptoms, namely familial hypocalciuric hypercalcemia (FHH), neonatal severe primary hyperparathyroidism (NSHPT), and neonatal hyperparathyroidism (NHPT) [2,3,4].

FHH comprises a genetically heterogeneous group [5]: FHH1 [OMIM #145980] is usually caused by heterozygous inactivating mutations in the calcium sensing receptor gene (*CaSR*) and accounts for about 65% of all FHH cases [3]. Mutations in other genes encoding for proteins with a role in calcium signaling (*GNA11* and *AP2S1*) cause FHH2 and 3 [6,7,8].

The aims of this review are to present the pathophysiology of calcium-sensing receptor signaling disorders with a focus on intra-uterine calcium homeostasis and the differing clinical outcomes in neonates.

## 2. Epidemiology

FHH1 is considered a rare disease. However, the prevalence of FHH is difficult to estimate since many of those affected remain asymptomatic and remain undetected. In 2002, Hinnie et al. estimated a minimal prevalence for FHH of 1 in 78,000 (1.3 cases in 100,000) [9].

However, in a recent report, Dershem et al., presented results from whole exome analyses in 51,289 probands of the DiscovEHR cohort in northern USA and detected a prevalence of *CaSR* mutations (giving rise to FHH1) of 74.1 per 100,000. The authors state that these results indicate a prevalence of FHH comparable to that of primary hyperparathyroidism (PHPT). If these findings can be confirmed, FHH cannot be considered a rare disease anymore [10].

In Germany, with an estimated livebirths of 780,000 per year, this prevalence would give rise to 580 neonates with an inherited form of FHH. Of these, about 50% will be affected by paternal inheritance and in addition, sporadic cases will occur. However, given the fact that NHPT is reported so rarely in the literature, these calculations are difficult to perform. [11].

## 3. Pathophysiology

FHH1 (OMIM: 145980) is caused loss-of-function (LOF) mutations in *CaSR* on chromosome 3q21.1. This is the most common form of FHH with more than 300 mutations in *CaSR* reported to date. The calcium-sensing receptor (*CaSR*) is a G protein-coupled transmembrane receptor. When activated, the receptor couples to heterotrimeric G proteins (Ga11) leading to an increased phospholipase C activity with a subsequent elevation of inositol 1,4,5-trisphosphate (IP3) and a rise in cytoplasmic ionized calcium [12,13] (Figure 1).

The level of plasma membrane expression is regulated by the balance of agonist-driven insertional signaling (ADIS) and the activity of adaptor-related protein complex 2 (AP2). ADIS provide trafficking of *CaSR* from the intracellular reservoir to the plasma membrane after sustained receptor activation [14]. AP2 together with β-arrestin represent the key regulators of *CaSR* internalization and facilitate clathrin-mediated endocytosis (Figure 1) [15].

Under physiological conditions, several organ systems are affected by activation of the *CaSR* due to hypercalcemia (Figure 2). In the parathyroid glands, activation of the *CaSR* results in suppression of PTH secretion [16,17]. In the kidneys, the *CaSR* is expressed throughout most of the nephron and stimulation of *CaSR* by Ca^2+^ leads to reduced urinary concentration capacity [18,19,20]. In addition, *CaSR* is expressed in both osteoblasts and osteoclasts. Stimulation of the *CaSR* by hypercalcemia leads to inhibition of osteoclastic resorption and increased osteoblastic activity [21,22,23]. However, these effects do not result in increased bone mass as indirectly concluded by the lack of BMD changes with calcium only supplementation or calcimimetic administration or in prolonged hypercalcemic states. Relevance of *CaSR* expression in other skeletal cells such as osteocytes or chondrocytes remains to be determined [24].

FHH2 (OMIM: 145981) is caused by inactivating mutations of the gene *GNA11* located on chromosome 19p, encoding the Gα11 protein [25,26]. LOF mutations in *GNA11* lead to disruption of the G-protein activation that impairs *CaSR* signal transduction [27]. FHH3 (OMIM: 600740) is caused by mutations of the gene *AP2S1* located on chromosome 19q13.3. *AP2S1* encodes the sigma subunit of the adaptor-related protein-2 (AP2σ), which is involved in clathrin-mediated *CaSR* endocytosis. The mutant AP2σ leads to enhanced *CaSR* cell-membrane expression but also impaired *CaSR* signaling [6,28].

LOF of *CaSR* (in FHH1) or its downstream partner signaling proteins (in FHH2 and FHH3) result in an elevation of the body’s calcium set point, defined as the serum calcium concentration at which PTH secretion is half-maximal. Patients with FHH display higher levels of plasma PTH and it takes a higher level of plasma calcium to suppress PTH secretion (Figure 3). The elevated PTH stimulates bone resorption resulting in elevated serum calcium levels. The resulting hypercalcemia would normally inhibit osteoclast activity, however, due to the LOF in the *CaSR*-pathway in osteoclasts, this mechanism does not work properly.

Further, *CaSR* is expressed in the nephron of the kidneys and in the LOF situation, the renal threshold for calcium is increased and results in low renal calcium excretion compared to the enhanced filtered load of calcium.

## 4. Diagnosis, Screening and Prevention

### 4.1. Clinical Signs and Symptoms 

Clinically, FHH is a benign, typically asymptomatic disease and usually presents with the biochemical triad: life-long, non-progressive hypercalcemia, normal or slightly increased serum PTH levels and hypocalciuria [2]. Serum phosphate levels are often reduced, circulating levels of 25OHD are reported as normal in FHH [2]^.^ However, as discussed below, vitamin D status may influence the phenotype of FHH [29]. 1,25-(OH)2D levels are normal or elevated [2,30] and mild hypermagnesemia may be present [5,25,31].

Accordingly, the majority of patients with FHH do not require medical or surgical treatment [5,32].

Although typically asymptomatic, some patients with moderate-to-severe hypercalcemia may present with classical hypercalcemic symptoms (i.e., polyuria- polydipsia, fatigue, acute and chronic pancreatitis, gallstones and chondrocalcinosis, acute pancreatitis, chondrocalcinosis, and nephrolithiasis) [5,33,34,35,36,37]. Although, PTH is elevated in FHH, patients generally have BMDs and Z-scores comparable to normal controls [2,38,39].

The phenotypes of the different subtypes of FHH are generally similar [5,26], aside from some reports of more symptomatic disease in FHH3 with more severe hypercalcemia, hypermagnesemia as well as more pronounced hypocalciuria. Low bone mineral density (BMD) and cognitive impairment have been reported in FHH3, aswell [15,40].

### 4.2. Neonatal Diseases

In contrast to the clinically benign course of FHH, neonates with de-novo or paternally derived mutations in *CaSR* may present with neonatal severe primary hyperparathyroidism (NSHPT). NSHPT [OMIM #239200] is a clinically severe, ultra-rare disease associated with a high mortality usually caused by homozygous or compound heterozygous inactivating mutations in *CaSR* [2,41]. Infants with NSHPT develop significant and symptomatic hypercalcemia with muscular weakness, respiratory distress, fractures, and failure to thrive in the early days of life [17,42,43]. Swift and rigorous interventions to decrease calcium levels and suppress PTH secretion are necessary to decrease mortality in this condition. Treatment is discussed below.

Milder phenotypes are termed neonatal hyperparathyroidism (NHPT). These conditions are characterized by elevated serum PTH levels and subsequent bone disease, however only moderate hypercalcemia. Infants with NHPT are typically symptomatic as hypercalcemia leads to poor feeding with resulting dehydration, and lethargy, while PTH excess causes skeletal demineralization. These patients usually carry heterozygous, LOF- mutations in *CaSR* [4].

The severity of the neonatal disease in infants with heterozygous LOF-*CaSR* mutations is not only defined by the underlying genetic mutation but is significantly modified by the parental origin of the mutation (Figure 4) [17,44]. 

When the mutation is maternally derived, both the mother and fetus share a need for increased calcium levels and coincide in the regulation of calcium homeostasis [11]. Thus, these pregnancies are likely uneventful. In contrast, when the mutation is paternally derived or sporadic (de novo), this result in differing calcium setpoints in mother and fetus with a different calcium need and thus conflicting regulation of calcium levels. In these pregnancies, the normocalcemic maternal environment is sensed as hypocalcemic by the fetus and thus induces fetal hyperparathyroidism to support increased fetal calcium levels at the expense of skeletal mineralization. This may result in bone fragility and pre- and perinatal fractures [11,44,45].

Accordingly, the majority of the reported cases of NHPT/NSHPT based on heterozygous mutations in *CaSR*, are neonates born to normocalcemic mothers with a paternally inherited or a de-novo mutation in *CaSR* [41,46,47,48]. The maternal vitamin D status during pregnancy may be a modifying factor in the phenotypic variability of heterozygous *CaSR* mutations [29,41,49]. Despite the severity of the phenotype, spontaneous recoveries are common. Wilkonson et al. reported on spontaneous clinical improvement in an infant with NHPT due to a paternally inherited heterozygous *CaSR* mutation [50]. In fact, spontaneous recovery should be expected in NHPT since after birth, the counter-regulating maternal environment is no longer influencing the fetal organism [45].

### 4.3. Establishing the Diagnosis

As the biochemical hallmarks of FHH are elevated calcium and PTH levels, the main differential diagnosis for FHH is primary hyperparathyroidism (PHPT), a condition that is ultra-rare in pediatric patients and has never been reported in neonates [51]. As such, in the neonatal setting the diagnostic challenge is to distinguish forms of NSHPT based on homozygous or compound heterozygous mutations in *CaSR*, which require aggressive and definite calcium and PTH suppressive measures, from milder forms such as NHPT which will resolve within the first months of life. Ultimately, the severity of the disease dictates the extent of treatment, and the differential diagnosis will be made based on molecular genetic findings [11].

In the clinical practice for older children (often adolescents) and adults, the differential diagnosis between primary hyperparathyroidism (PHPT) and FHH is essential, as parathyroidectomy may be indicated in PHPT and in neonates with NSHPT but usually not in older patients with FHH [32,52,53].

Recommendations on the practical management of parathyroid disorders is provided in the recently published European expert consensus paper [53].

In cases of chronic hypercalcemia, it is of utmost importance to measure calcium excretion in the urine. Neither the calcium level nor the PTH level allows a differentiation of FHH and PHPT [53]. However, more than 80% of FHH-affected individuals present with hypocalciuria (Ca creatinine clearance ratio [CCCR] < 0.01) in the presence of hypercalcemia, whereas less than 20% of patients with PHPT present with a CCCR < 0.01 [54,55]. In general, a CCCR less than 0.01 is considered to proof FHH [56,57]. However, a CCCR between 0.01 and 0.02 has been reported in some patients with FHH and Christensen et al. found a diagnostic sensitivity for a threshold of 0.02 [58].

In 2018, Bertocchio et al. introduced the pro-FHH (probability of having FHH) score with a specificity for FHH and PHPT of 100%. Pro-FHH takes plasma calcium, PTH, and serum osteocalcin concentrations, and CCCR calculated from 24-h urine collection into account [59]. However, the pro-FHH score has only been evaluated in adults and interpretation has been shown to be difficult in patients with elevated PTH levels [56].

Vargas-Poussou et al. compared a large cohort of FHH and PHPT patients and found considerable overlap in the phenotypes, making the distinction based on laboratory data alone difficult [60]. Thus, Christensen et al. suggest a two-step diagnostic procedure, where the first step is based on the CCCR with a cut-off at <0.020, and the second step is *CaSR* gene analysis. The clinical presentation in FHH1 and FHH2 is usually similar. Mutation analysis of AP2S1 gene for FHH3 should be performed in FHH patients with marked hypermagnesemia, cognitive impairment and low bone mineral density [15].

## 5. Management

A specific treatment is not necessary in the majority of patients. In contrast, morbidity may result from inappropriate surgical intervention [5]. The prognosis in FHH is good, and the life expectancy is probably normal [2].

However, in rare instances when calcium levels are very high or the patients become symptomatic from hypercalcemia, it becomes necessary to treat FHH [11,61].

Calcimimetic drugs, such as cinacalcet, are allosteric agonists at the *CaSR* [61]. They enhance the effect of extracellular calcium at the *CaSR* in the parathyroid cell, thereby decreasing PTH secretion and consequently serum calcium levels. In FHH, cinacalcet results in a significant reduction of serum calcium levels and improvement of calcium clearance after initiation of cinacalcet therapy [17,61,62,63,64,65,66,67].

Treatment with cinacalcet has also been reported in FHH2 [7] and FHH3 [68], with successful normalization of serum calcium concentrations. In addition, infants with NHPT and NSHPT have also been treated successfully with cinacalcet [11,17,42]. However, in NHPT it remains unclear, whether the use of cinacalcet is necessary or whether surveillance alone is sufficient to confirm a spontaneous improvement over time. In NSHPT, total parathyroidectomy is generally considered the definitive treatment. Subtotal parathyroidectomy is ineffective and therefore not recommended [69,70]. Prior to surgery, serum calcium levels need to be lowered. For treatment of hypercalcemia bisphosphonates are well established and their use has been reported in neonates with NSHPT and NHPT [67].

## 6. Genetic Counselling and Neonatal Management

Based on the fact that most patients with FHH do not require any form of treatment and that patients are at risk of unnecessary (partial) parathyroidectomies the focus is based on (genetic) counselling [53]. Perinatal complications in pregnancies affected by FHH should also be brought to the attention of affected patients of reproductive age and to the clinicians potentially involved in their care, e.g., human geneticists, obstetricians, (pediatric) endocrinologists, and neonatologists. The fact that the paternally derived mutations cause a greater risk for neonates to suffer from hyperparathyroid complications emphasizes the need to specifically instruct male patients with FHH on these potential complications.

Therefore, we agree with the recommendation of Ghaznavi et al., that it is advisable to offer genetic counseling to all pregnant women with confirmed FHH or a partner with FHH given the range of possible neonatal outcomes [52], including NSHPT in neonates with homozygous or compound heterozygous inactivating mutations [32].

## 7. Summary

While FHH in general is a benign condition, neonates with inactivating mutations in *CaSR* are at risk to develop NHPT or even NSHPT. This risk is highest if the mother is not affected by FHH or if homozygous or compound heterozygous mutations affect the neonate. In NHPT clinical signs are often temporary and resolve within weeks or months once the neonate has established calcium levels according to her/his individual threshold. At that point, the excess PTH will drop to the individual threshold and the skeletal system may mineralize appropriately in the further course. However, in NSHPT (due to homozygous or compound heterozygous mutations in *CaSR*), therapeutic interventions must aim to lower calcium and PTH levels until a diagnosis is established and a definite (surgical) treatment can be pursued. Adult patients with FHH of reproductive age should be counselled about the potential peripartal complications in this condition.

## Figures and Tables

**Figure 1 jcm-11-02595-f001:**
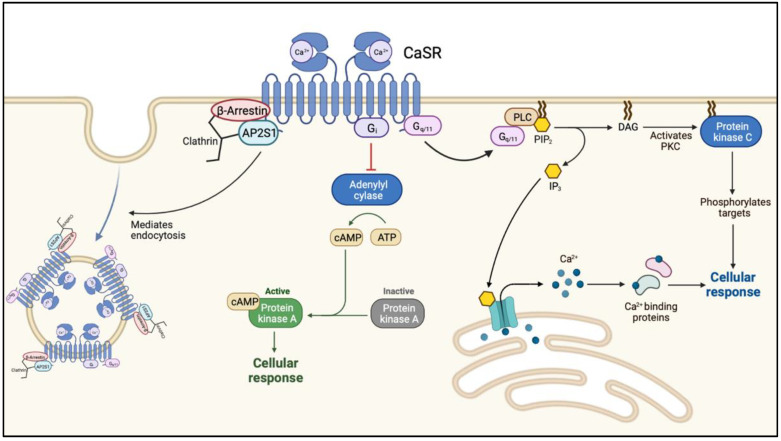
The intracellular cascade following ligand (calcium-) binding to the calcium-sensing receptor. Abbreviations: Ca^2+^, calcium; *CaSR*, calcium sensing receptor; AP2S1, adaptor-related protein complex 2 subunit 2; G_i_, inhibitory G-protein; cAMP, cyclic adenosine monophosphate; PLC, phospholipase C; PIP3, phosphatidylinositol-3,4,5-triphosphate; DAG, diacylglycerol; PKC, protein kinase C.

**Figure 2 jcm-11-02595-f002:**
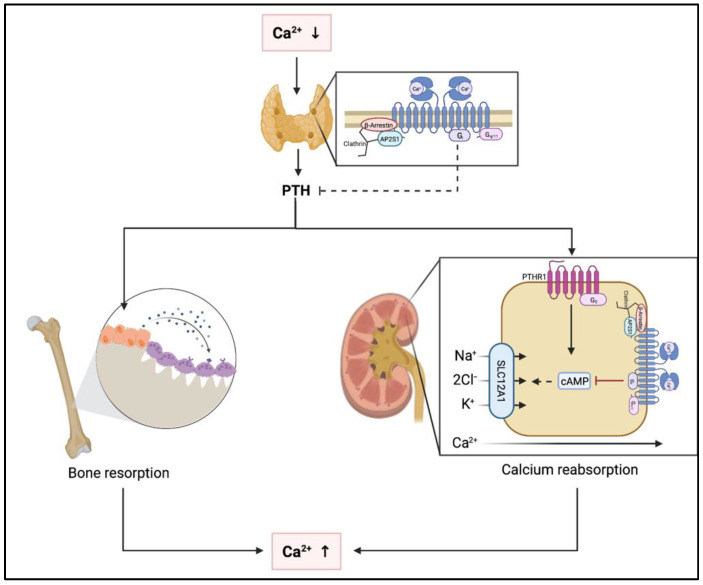
Pathophysiological effects of hypo- and hypercalcemia at the parathyroid glands, the kidneys and the skeleton. Abbreviations: Ca^2+^, calcium; PTH, parathyroid hormone; PTHR1, parathyroid hormone receptor 1; cAMP, cyclic adenosine monophosphate; SLC1A1, solute carrier 1A1; Na^+^, sodium; Cl^−^, chloride; K^+^, potassium.

**Figure 3 jcm-11-02595-f003:**
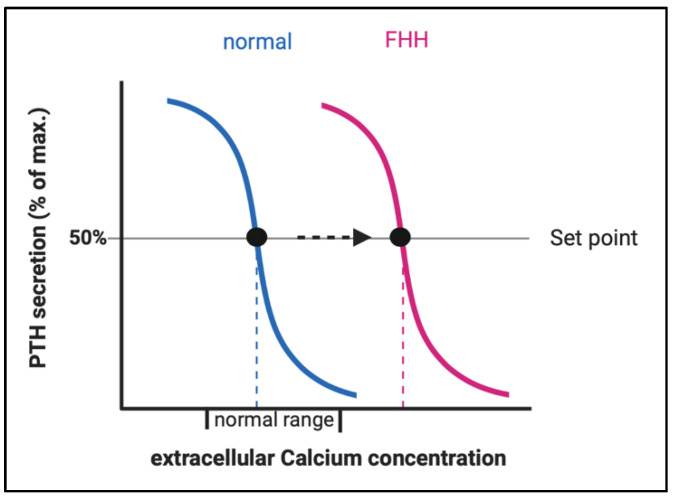
Change of the set point of the calcium concentration for half maximal PTH secretion in FHH. Abbreviations: FHH, familial hypocalciuric hypercalcemia.

**Figure 4 jcm-11-02595-f004:**
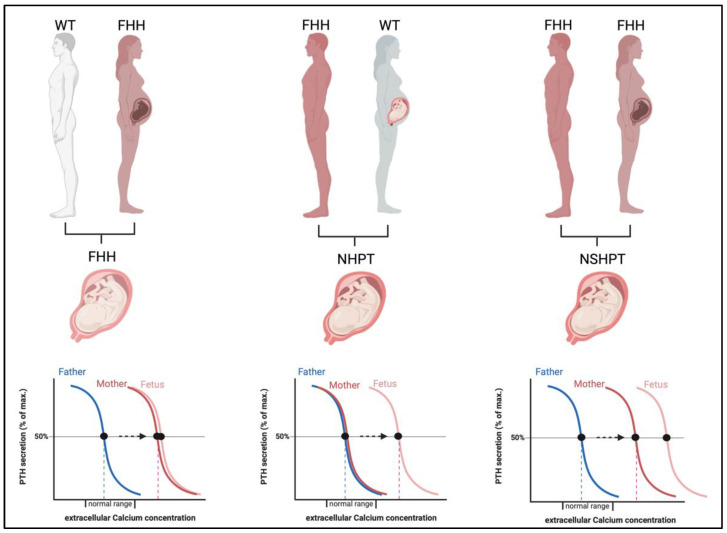
Impact of the in-utero environment on fetal calcium homeostasis in calcium-hyposensitivity disorders. If the fetus is affected by heterozygous *CaSR* mutations, fetal and maternal calcium needs are concordant (**left panel**). If the maternal environment is normocalcemic (**middle panel**) but the fetus affected by a paternally derived or de-novo mutation in *CaSR*, the maternal environment is perceived as hypocalcemic by the fetus and PTH stimulation arises resulting in NHPT. If both parents are affected by FHH (**right panel**), the fetus may inherit both mutations and a most severe stimulation of PTH and hypercalcemia arises, a NSHPT. Abbreviations: FHH, familial hypocalciuric hypercalcemia; WT, wild type; NHPT, neonatal hyperparathyroidism; NSHPT, neonatal severe primary hyperparathyroidism.

## Data Availability

Not applicable.

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
