# Peer review of "Disorders of the Calcium Sensing Signaling Pathway: From Familial Hypocalciuric Hypercalcemia (FHH) to Life Threatening Conditions in Infancy"

_jcm, 2022, doi:10.3390/jcm11092595_

Round 1
Reviewer 1 Report
In the present review Höppner et al analyze the genetics, pathophysiology and management of disorders of the calcium sensing signaling pathway. The article is interesting and well-written
Comments
- Are there differences in presentation/diagnosis between sporadic cases of CaSR sensing or signaling reduction and genetic ones?
- Page 3 – lines 77-78: “Stimulation of the CaSR by hypercalcemia leads to inhibition of osteoclastic resorption and increased osteoblastic activity”. However, these effects do not result in increased bone mass as indirectly concluded by the lack of BMD changes with calcium only supplementation or calcimimetic administration or in prolonged hypercalcemic states
- In section 4.1 please quote changes in other parameters (serum and urinary phosphate, Mg, 25OHD, 1,25(OH)2D)
- Do patients with FHH have reduced bone mass or increased fracture risk due to chronically increased PTH?
- What is the mechanism of hypermagnesemia in FHH3?
Minor comments
- Abstract – page 1 – line 20: “conditions” instead of “a condition”
- Page 1 – line 28: “this is based” instead of “this based”
- Line 194: “FHH and PHPT” instead of “FHH to PHPT”?
Author Response
We thank the Reviewers for the review of our manuscript and the helpful comments and suggestions. Below we have answered in a point-by-point response. The requested changes are highlighted in the manuscript and are copied into the point-by-point rebuttal for easier referral.

Reviewer 2 Report
GENERAL COMMENT
The authors reviewed the literature on FHH, focusing on neonatal severe hyperparathyroidism (NSHPT)
I suggest to add the aims at the end of Introduction section
The paper is a narrative review, nevertheless, I recommend adding a brief Methods section in which you should report the period of inclusion, the databases searched, and the selection criteria.
Lines 223: I recommend to point out that the thyroid gland must be spared (10.1016/j.ijsu.2014.05.042)
The author should add the meaning of all acronyms used in the figures
Figure 4: I suggest to improve the quality of graphs
The authors should complete the sections under the Summary
Author Response

(The authors gave the same response as above.)
